# Explicitly Bounding Q-Function Estimates for Offline-to-Online Reinforcement Learning

## Abstract

Offline-to-Online Reinforcement Learning (O2O RL) presents a compelling framework for deploying decision-making agents in domains where online data collection is limited by practical constraints such as cost, risk, or latency. In this paradigm, agents are initially trained on fixed datasets and subsequently refined through limited online interaction. However, this transition exposes a fundamental challenge: the misestimation of state-action values associated with out-of-distribution (OOD) actions, inherited from the offline phase. Such misestimations can severely destabilize online adaptation, leading to suboptimal policy behavior. To address this, we propose an algorithm-agnostic method that regularizes the Q-function prior to fine-tuning by injecting structured noise into dataset actions. This process explicitly bounds Q-value estimates across the entire action space, not just in-distribution actions, mitigating both overestimation and underestimation. We introduce a tunable parameter that governs the degree of conservatism and optimism within the Q-value bounds during the online fine-tuning phase. Extensive empirical evaluations on standard O2O RL benchmarks demonstrate that our method yields substantial improvements over strong baselines, both in terms of stability and final performance. These results underscore the importance of principled Q-function initialization and offer a practical path toward more robust reinforcement learning under distributional shift.

## 1 Introduction

Offline Reinforcement Learning (RL) is a promising paradigm for training decision-making agents in domains where real-world interaction is expensive, time-consuming, or potentially hazardous. Applications such as robotic manipulation (Chebotar et al., 2021; Bhateja et al., 2023), autonomous driving (Codevilla et al., 2019; Shi et al., 2021), and healthcare (Gottesman et al., 2019; Tang et al., 2022) exemplify high-stakes environments where collecting interactive data is costly or risky, making conventional online RL approaches impractical. By leveraging static datasets, offline RL eliminates the need for online data collection. In many real-world scenarios, agents retain the ability to interact with the environment after initial training, enabling further policy improvement. This motivates the Offline-to-Online RL (O2O RL) paradigm, which fine-tunes offline-trained policies through online updates, combining sample efficiency with adaptability. Still, this transition introduces several challenges that can hinder effective policy refinement.

A key challenge in O2O RL is the distributional mismatch between the offline dataset and the online exploration policy. Since offline RL is limited to the support of a fixed dataset, it is particularly vulnerable to out-of-distribution (OOD) action misestimation, where the learned value function assigns inaccurately high or low values to actions that are poorly represented in the offline data (Fujimoto et al., 2019). These misestimated values can drive the agent toward suboptimal or unstable actions during online fine-tuning, potentially degrading performance and necessitating additional recovery steps. In particular, underestimated Q-values may propagate through bootstrapping, impeding effective learning (Kumar et al., 2019; Zhou et al., 2024). At the same time, extrapolation errors introduced by evaluating poorly supported actions can lead to severe overestimation (Levine et al., 2020), destabilizing the learning process.

To mitigate such problems, prior work has proposed calibration techniques that aim to reduce Q-value underestimation and prevent sudden performance drops during the early stages of online adap-

tation (Nakamoto et al., 2023; Zhang et al., 2024). However, these methods often rely on specific offline RL algorithms or remain sensitive to the initialization of Q-values. Other approaches seek to combine offline datasets with online rollouts to minimize distributional mismatch and improve sample efficiency (Shin et al., 2025). However, these algorithms typically require a substantial amount of online sampling. Zhou et al. (2024) demonstrated that collecting online samples using a frozen pretrained policy, without retaining the offline data during fine-tuning, can mitigate performance degradation. Yet, this method also exhibits sensitivity to the initial Q-values, which can lead to significant performance variability.

We approach this problem from a different perspective by answering the following question: *How can Q-values be initialized in an offline RL algorithm-agnostic manner that jointly mitigates both underestimation and overestimation before online finetuning begins?* To address this, we propose a simple yet effective algorithm-agnostic method, **Bounding Q-function Estimates for Offline to Online RL (BOTO)**, which injects action noise during Q-learning prior to online finetuning. Crucially, this process produces a bounded Q-function that explicitly regularizes value estimates across the entire action space, covering not only in-distribution actions but also out-of-distribution ones. By enabling control over the degree of conservatism and optimism within this bound, BOTO helps prevent extreme estimation errors that commonly destabilize early online learning. We show that this strategy enhances both the stability and final performance of O2O RL. Our main contributions are as follows:

- We introduce a Q-value bounding method via action noise injection that explicitly regularizes OOD actions and balances underestimation and overestimation.
- We provide an analysis showing that our method corresponds to Q-learning in an $\alpha$-Noisy Action MDP, and derive bounds that characterize its impact on Q-value estimates across the action space.
- We perform experiments on standard O2O RL benchmarks, demonstrating improvements in policy performance over existing methods.

To support these contributions, we first revisit the foundations of offline and online reinforcement learning to identify how Q-value misestimation emerges in the O2O RL setting. We then analyze the mechanisms through which underestimation and overestimation affect policy learning during fine-tuning. Building on this analysis, we introduce an approach to bounding Q-function estimation based on action noise injection and provide a detailed examination of its properties. Finally, we demonstrate the effectiveness of our method through empirical evaluations on standard O2O RL benchmarks.

## 2 BACKGROUNDS AND RELATED WORK

Reinforcement learning (RL) aims to find an optimal policy for sequential decision-making tasks, which are typically formalized as Markov Decision Processes (MDPs), defined by $(\mathcal{S}, \mathcal{A}, R, P, \gamma)$ (Sutton et al., 1998). Here, we focus on control settings, and accordingly assume that the state space $\mathcal{S} \in \mathbb{R}^n$ and the action space $\mathcal{A} \in \mathbb{R}^m$. $R$ is the reward function, $P$ represents the transition dynamics and $\gamma \in (0, 1)$ is the discount factor. The objective is to learn a policy $\pi(a \mid s)$ that maximizes the expected return $\eta(\pi) = \mathbb{E}_\pi \left[ \sum_{t=0}^{\infty} \gamma^t r_t \right]$, where $r_t$ is the reward obtained at timestep $t$. This is generally done by iteratively improving the policy to approach the optimal policy $\pi^* = \arg\max_\pi \eta(\pi)$. The action-value function (or Q-function) associated with a policy $\pi$ is defined as $Q^\pi(s, a) = \mathbb{E}_\pi \left[ \sum_{t=0}^{\infty} \gamma^t r_t \mid s_0 = s, a_0 = a \right]$, which quantifies the expected discounted return from state $s$, taking action $a$, and then following policy $\pi$ thereafter.

**Offline-to-Online RL.** Offline RL aims to learn a policy exclusively from a fixed dataset, without any additional interaction with the environment (Levine et al., 2020; Prudencio et al., 2023). In contrast, online RL continuously updates the policy through active interaction with the environment. The concept of Offline-to-Online Reinforcement Learning (O2O RL), which leverages a pretrained offline policy and value function for subsequent online fine-tuning, is inspired by a common paradigm in modern machine learning: models are first pre-trained on large-scale, general-purpose datasets and then fine-tuned on smaller datasets to achieve optimal performance in the target domain (Nakamoto et al., 2023).

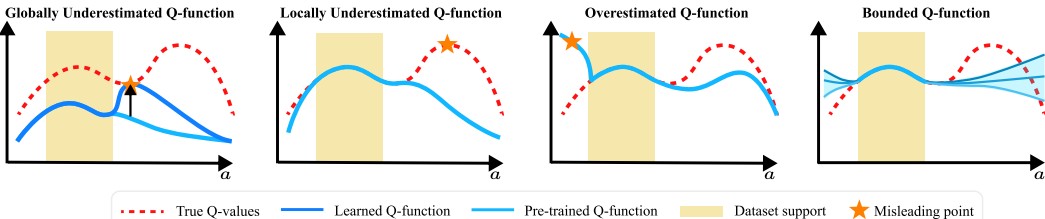

Figure 1: Illustration of Q-value misestimation cases in O2O RL. Global underestimation favors suboptimal actions. Local underestimation suppresses truly high-value actions. Overestimation misleads the policy toward spurious actions. Bounding Q-values on OOD actions mitigates both extremes, enabling stable estimates. Our method adjusts the level of optimism or conservatism within this bound.

Despite its promise, O2O RL faces several challenges, primarily due to the distributional shift between offline and online data, as well as inaccuracies in the value function for OOD state-action pairs, which arise from training the target policy using a limited offline dataset. Zhou et al. (2024) identify two key issues arising from these challenges: unlearning, a temporary performance drop at the onset of fine-tuning requiring several recovery steps, and forgetting, where early fine-tuning erodes the pretrained initialization, making recovery through online learning nearly impossible.

To mitigate such problems, prior work has explored strategies. Calibration methods address Q-value underestimation and reduce performance degradation during online adaptation. Nakamoto et al. (2023) rescale conservative Q-values to balance caution and adaptability, while Guo et al. (2023) use uncertainty-aware estimation to stabilize early updates. Zhang et al. (2024) employ ensemble critics to suppress misestimation and ensure stable fine-tuning. Other methods reduce distributional mismatch by combining offline and online data. Ji et al. (2023) blend data using a confidence-aware schedule, and Shin et al. (2025) apply prioritized replay to focus on high-quality offline samples. Zhang et al. (2024) also alternate updates to maintain distributional continuity, while Zhou et al. (2024) stress a high update-to-data ratio and interleaving offline data to prevent forgetting.

However, since these methods rely on the initial Q-values before online fine-tuning, they either suffer from unlearning at the early stage of fine-tuning or require a prolonged period to recover performance. To address this limitation, we propose a simple yet effective algorithm-agnostic strategy that improves the robustness of offline-to-online adaptation by mitigating Q-value misestimation throughout fine-tuning.

## 3 Q-VALUE ERROR IN O2O RL

Q-value misestimation is one of the key challenges in Offline-to-Online Reinforcement Learning (O2O RL). In standard deep RL frameworks, the policy network is trained to select actions that are assigned high values by the learned Q-function. In purely online settings, even when the Q-function is initially inaccurate, these errors can often be corrected over time: overestimated actions are executed and revised through feedback, while underestimated actions may eventually be explored due to the agent's inherent exploration strategy. This iterative process enables value estimation errors to diminish progressively.

In contrast, O2O RL relies on a Q-function pre-trained on static offline datasets with limited action coverage. The restricted support of these datasets often leads to inaccurate value estimates, particularly for out-of-distribution (OOD) actions. Consequently, during the early stages of online fine-tuning, the policy may act suboptimally or even degrade in performance due to its reliance on these misestimated Q-values, as observed by Nakamoto et al. (2023). To provide a deeper understanding of these failure modes, we now conduct a detailed analysis of Q-value underestimation and overestimation in the O2O RL setting. In Figure 1, we illustrate the failure modes associated with Q-value misestimation in O2O RL.

**Q-value underestimation** in O2O RL typically arises in two forms. Global underestimation is characteristic of approaches such as Conservative Q-Learning (Kumar et al., 2020), which impose conservative penalties to discourage the selection of OOD actions in offline settings. While this

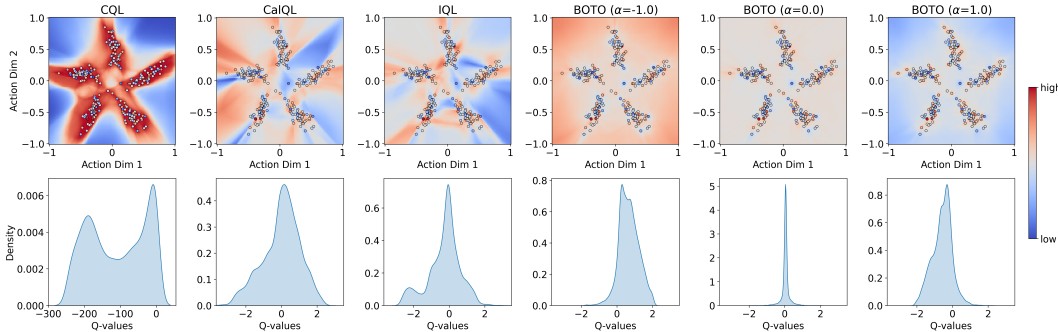

Figure 2: Comparison of Q-value distributions between BOTO and baseline offline RL algorithms (bandit setting). The top row visualizes Q-value estimates for sampled actions, where circles denote actions from the dataset and are colored according to their associated rewards. The bottom row presents the corresponding kernel density estimates (KDEs), highlighting differences in Q-value estimation behavior across methods.

promotes safety and stability during offline training, it can hinder policy improvement during online fine-tuning. Nakamoto et al. (2023) demonstrate that global underestimation can cause the policy to overlook optimal actions, instead favoring suboptimal alternatives due to suppressed Q-values. This misguidance is especially detrimental during the early stages of online adaptation, when accurate value estimates are critical for effective policy improvement. Local underestimation, on the other hand, occurs when a small subset of promising actions are assigned disproportionately low Q-values. As a result, the policy tends to avoid these underestimated actions, even if they are actually high quality. This leads it to repeatedly select suboptimal actions, which limits its capacity for improvement during online adaptation.

**Q-value overestimation** has been widely recognized as a core issue in offline RL, particularly because Q-functions are trained only on actions present in the limited offline dataset, leading to unreliable estimates for OOD actions (Van Hasselt et al., 2016). However, this issue is often mitigated during offline training through policy constraints or behavior regularization, which encourage the learned policy to stay close to the data distribution (Wu et al., 2019; Fujimoto & Gu, 2021; Tarasov et al., 2023). As a result, even if the Q-values for OOD actions are overestimated, the policy typically avoids selecting them, reducing their impact in practice.

The situation changes during online finetuning. When strong behavior regularization is retained, the policy may become overly conservative, limiting its ability to explore beyond the offline dataset. In contrast, relaxing this regularization allows the policy to exploit overestimated Q-values, potentially causing it to favor actions that appear optimal under the misestimated Q-function but are actually suboptimal. This can lead to unstable policy updates until the Q-values are sufficiently corrected. While moderate Q-value overestimation may encourage useful exploration, excessive overestimation often demands many update steps to correct, slowing down adaptation and degrading performance. Our approach addresses these issues by bounding the Q-function on OOD data, providing a reliable initialization for subsequent fine-tuning. This mechanism helps prevent sudden drops in policy performance during the early stages of adaptation.

## 4 BOUNDING Q-FUNCTION ESTIMATION

In the previous section, we discussed how Q-value underestimation and overestimation can substantially degrade policy performance during the O2O adaptation process. This challenge raises a central research question:

> *How can we initialize Q-values to support*
> *stable and effective fine-tuning of a policy pretrained on offline data?*

To address this, we introduce a simple yet effective approach that calibrates Q-values during the transition from offline pretraining to online fine-tuning. Drawing inspiration from Oh & Lee (2025), our approach, **Bounding Q-function Estimates for O2O RL (BOTO)**, introduces Q-value bounds

over the entire action space by dynamically adjusting target values, penalizing or rewarding them based on the magnitude of action perturbations. Furthermore, a tunable bias controller is incorporated to adjust the degree of conservatism or optimism within these bounds, enabling more flexible and robust Q-function initialization for fast adaptation during online fine-tuning.

To illustrate this effect, Figure 2 presents the estimated Q-values and corresponding kernel density estimates (KDEs) produced by our method under varying bias control parameters, alongside those from representative offline RL algorithms such as CQL (Kumar et al., 2020), CalQL (Nakamoto et al., 2023) and IQL (Kostrikov et al., 2021), all of which are commonly employed during the pre-training phase in prior works. These results are evaluated on action samples drawn from a Pinwheel distribution. Baseline approaches often exhibit pronounced Q-value underestimation or overestimation, which can significantly impair performance during subsequent fine-tuning. In contrast, our method produces Q-values that are consistently bounded across the entire action space, reducing both overestimation and underestimation, for out-of-distribution actions.

We begin by formalizing the bounding effect induced by injecting noise into the action space during Q-learning. Specifically, we minimize a regression loss over perturbed actions $\bar{a} \sim q_\sigma(\cdot \mid a)$, where $q_\sigma$ is a noise distribution parameterized by an action $a \in \mathcal{A}$ and a noise scale $\sigma > 0$, centered at $a$ with support satisfying $\mathcal{A} \subseteq \mathrm{supp}(q_\sigma)$. The target value consists of the original reward and a regularization term weighted by the deviation between $a$ and $\bar{a}$, encouraging the Q-function to generalize smoothly over perturbed actions. Formally, we define the following objective:

$$J(\theta) = \mathbb{E}_{\substack{(s,a,s') \sim \mathcal{D} \\ \bar{a} \sim q_\sigma(\cdot \mid a)}} \left[ (Q_\theta(s, \bar{a}) - \bar{y}(s, a, s', \bar{a}))^2 \right] \tag{1}$$

where the target $\bar{y}$ is defined as:

$$\bar{y}(s, a, s', \bar{a}) = y(s, a, s') - \alpha \|a - \bar{a}\|_2^2 \tag{2}$$

The variable $y$ represents the Bellman target, which varies by algorithm. Injecting noise ensures all actions have non-zero probability under the noise distribution. When $\alpha = 1$, our objective aligns with the Noisy Action MDP (NAMDP) from Oh & Lee (2025), where a penalty term in the reward suppresses Q-value overestimation for OOD actions. In this case, Q-learning with noisy actions effectively estimates the Q-function of a modified MDP.

We generalize this formulation by introducing $\alpha$ as a tunable parameter that modulates the degree of regularization applied to the reward function. $\alpha$ serves as a bias-control mechanism, allowing the learned Q-function to interpolate between conservative (underestimating) and optimistic (overestimating) estimates for OOD actions. This extension yields the $\alpha$-Noisy Action MDP ($\alpha$-NAMDP), a generalization of the original NAMDP framework. We further show that minimizing the objective in Equation (1) is equivalent to performing Q-learning in the $\alpha$-NAMDP setting.

**Definition 1** ($\alpha$-Noisy Action MDP). *Given a noise distribution $q_\sigma$, a finite dataset $\mathcal{D} = \{(s_i, a_i, r_i, s_i')\}_i$ and a dataset distribution $p_\mathcal{D}$, the NAMDP is defined as an MDP $(\mathcal{S}, \mathcal{A}, R_\sigma, P_\sigma, \gamma)$, where:*

$$P_\sigma(s' \mid s, \bar{a}) = \int_\mathcal{A} p_\mathcal{D}(s' \mid s, a) \, p_\mathcal{D}(\bar{a} \mid s, a, \sigma) \, da,$$

$$R_\sigma(s, \bar{a}) = \int_\mathcal{A} p_\mathcal{D}(\bar{a} \mid s, a, \sigma) \left( R(s, a) - \alpha \|a - \bar{a}\|_2^2 \right) da.$$

*Here, $p_\mathcal{D}$ denotes the empirical dataset distribution induced by the offline dataset $\mathcal{D}$. The conditional distribution $p_\mathcal{D}(\bar{a} \mid s, a, \sigma)$ is defined as:*

$$p_\mathcal{D}(\bar{a} \mid s, a, \sigma) = \frac{p_\mathcal{D}(a \mid s) \, q_\sigma(\bar{a} \mid a)}{\int_\mathcal{A} p_\mathcal{D}(\tilde{a} \mid s) \, q_\sigma(\bar{a} \mid \tilde{a}) \, d\tilde{a}}.$$

We now establish that minimizing the proposed learning objective in Equation (1) is equivalent to performing Q-learning under the dynamic of the $\alpha$-NAMDP. The equivalence is formalized in the following theorem:

**Theorem 1** ($\alpha$-NAMDP Objective). *Given that the function $Q$ minimizes the following objective:*

$$J(Q) = \mathbb{E}_{a \sim p_\mathcal{D}(\cdot \mid s), s' \sim p_\mathcal{D}(\cdot \mid s, a), \bar{a} \sim q_\sigma(\cdot \mid a)} \left[ (Q(s, \bar{a}) - \bar{y}(s, a, s', \bar{a}))^2 \right], \tag{3}$$

*where the target value $\bar{y}(s, a, \bar{a})$ is defined as:*

$$\bar{y}(s, a, s', \bar{a}) = \mathbb{E}_{\tilde{a} \sim \pi(\cdot | s')} \left[ R(s, a) + \gamma Q^\pi(s', \tilde{a}) - \alpha \|a - \bar{a}\|_2^2 \right],$$

*Then, the function Q is the Q-value function of $\pi$ in the $\alpha$-NAMDP.*

This theorem clarifies the role of the parameter $\alpha$ in shaping the learned Q-function. Specifically, it establishes that the Q-function optimized under our objective corresponds to the value function of a policy within the dynamics and reward structure defined by the $\alpha$-NAMDP. Consequently, $\alpha$ modulates the trade-off between conservative and optimistic value estimates for OOD actions.

Building on this result, we examine how varying values of $\alpha$ influence the learned Q-values, particularly in terms of their alignment with the ground-truth Q-values under the $\alpha$-NAMDP. As shown in Figure 3, negative $\alpha$ values induce overestimation for OOD actions, while positive values lead to underestimation. Furthermore, the magnitude of deviation grows with $|\alpha|$, amplifying the estimation bias in both directions. This behavior highlights how our objective systematically shapes the learned Q-function as a function of $\alpha$.

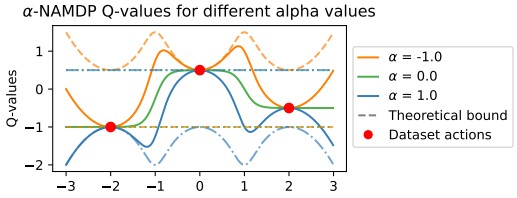

We further show that the Q-function learned under our objective is provably bounded over the entire action space, including OOD actions. Standard offline RL algorithms are trained only on dataset-supported actions and do not offer such guarantees for OOD actions unless additional, often strong, assumptions are made about the value function class. As a result, they may produce Q-functions with severely misestimated values for

Figure 3: Ground-truth Q-values and bounds under different $\alpha$ values in the $\alpha$-NAMDP (badint setting). Each curve shows the learned Q-values for a different $\alpha$ using $q_\sigma(\cdot \mid a) = \mathcal{N}(a, 0.25^2)$. Red dots represent the ground-truth Q-values of dataset actions under the original MDP. Dashed lines indicate the bounds derived in Theorem 2.

unseen actions. In contrast, our formulation enables controlled modulation of Q-value estimates across the entire action space through $\alpha$, contributing to more reliable online adaptation. The following theorem formalizes this boundedness:

**Theorem 2** (Q-value bound in the $\alpha$-NAMDP). *Let $\theta^*$ denote the parameters of $Q_\theta$ that minimize the functional objective defined in Equation (3), assuming a sufficiently expressive parameterization of the Q-function. Let $V^\pi$ be the true state value function under policy $\pi$ in the $\alpha$-NAMDP. Then, for any policy $\pi$, the learned Q-function $Q_{\theta^*}$ satisfies:*

$$L^\pi(s, a) \le Q_{\theta^*}(s, a) \le U^\pi(s, a) \quad \forall(s, a) \in \mathcal{S} \times \mathcal{A}$$

*where*

$$L^\pi(s, a) = \min(0, -\alpha) \cdot \mathcal{R}(s, a) + \inf_{a \in supp(p_D(\cdot|s))} \left[ R(s, a) + \gamma \int_{\mathcal{S}} p_D(s' \mid s, a) V^\pi(s') \, ds' \right]$$

$$U^\pi(s, a) = \max(0, -\alpha) \cdot \mathcal{R}(s, a) + \sup_{a \in supp(p_D(\cdot|s))} \left[ R(s, a) + \gamma \int_{\mathcal{S}} p_D(s' \mid s, a) V^\pi(s') \, ds' \right].$$

Here, $\mathcal{R}(s, a) = \int_{\mathcal{A}} p_D(\bar{a} \mid s, a, \sigma) \|a - \bar{a}\|_2^2 \, da$. We illustrate the Q-value bounds in the $\alpha$-NAMDP through an example in Figure 3, and provide the corresponding of the theorems proofs in the Section A.

### 4.1 METHOD

We propose an algorithm-agnostic method, Bounding Q-function Estimates for Offline to Online RL (BOTO), designed to mitigate both overestimation and underestimation in Q-value learning by introducing a regulated optimism phase. This phase adjusts Q-values before online fine-tuning, effectively managing out-of-distribution (OOD) actions. The level of optimism or conservatism is modulated via a bias-control parameter $\alpha$. This influences the Q-function's behavior particularly on actions not well-represented in the offline data. BOTO consists of three key stages.

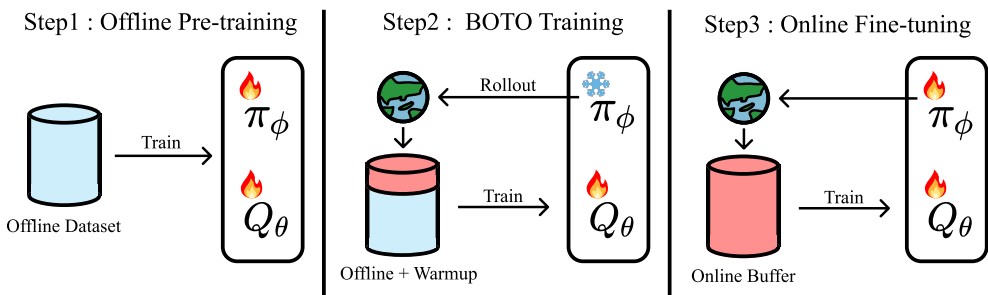

Figure 4: Overview of the Bounding Q-function Estimates for O2O RL (BOTO) framework. Our method comprises three sequential stages: (1) Offline Pre-training: An offline policy $\pi$ and Q-function $Q$ are learned using a fixed offline dataset via an offline RL algorithm. (2) BOTO Training: The pretrained Q-function is further optimized using the BOTO objective with the bias-control parameter $\alpha$, leveraging both the offline dataset and the warmup samples collected by the frozen pretrained policy. (3) Online Fine-tuning: Both the policy and the Q-function are fine-tuned using an online RL algorithm to enable stable and sample-efficient adaptation during the online phase.

**Offline Pre-training**   We first pretrain the policy and Q-function using a standard offline reinforcement learning algorithm on a fixed dataset $\mathcal{D}_{\text{offline}}$. This initializes the networks for effective warm-up and fine-tuning phases.

**BOTO Training**   We further optimize the pretrained Q-function $Q_\theta$ and policy $\pi_\phi$ using the objective defined in Equation (1). To reduce the distributional mismatch between offline and online data distribution, we introduce a short warm-up phase during which additional data is collected using the pretrained policy $\pi_\phi$. The resulting dataset is denoted as $\mathcal{D}_{\text{warmup}}$. We then construct an augmented dataset, $\mathcal{D}_{\text{BOTO}} = \mathcal{D}_{\text{offline}} \cup \mathcal{D}_{\text{warmup}}$, which serves to bridge the gap between offline pretraining and online fine-tuning. Using $\mathcal{D}_{\text{BOTO}}$, we optimize the Q-function with the following:

$$J(\theta) = \mathbb{E}_{(s,a,s') \sim \mathcal{D}_{\text{BOTO}}, \bar{a} \sim q_\sigma(\cdot|a)} \left[ (Q_\theta(s, \bar{a}) - \bar{y}(s, a, s', \bar{a}))^2 \right]$$

The target value $\bar{y}$ incorporates the entropy term from Haarnoja et al. (2018) and is computed as:

$$\bar{y}(s, a, s', \bar{a}) = r(s, a) - \alpha \|a - \bar{a}\|_2^2 + \mathbb{E}_{\tilde{a} \sim \pi_\phi} \left[ \gamma Q_{\theta'}(s', \bar{a}) + \tau \log \pi_\phi(\tilde{a} \mid s) \right]$$

where $\pi_\phi$ is the policy network pretrained in the offline phase, $Q_{\theta'}$ is the target Q-network and $\tau$ is the entropy temperature.

**Online Fine-Tuning**   Finally, we jointly fine-tune the Q-function $Q_\theta$ and the policy $\pi_\phi$ using a standard online RL algorithm, providing a strong initialization for subsequent learning. The entire pipeline is visualized in Figure 4. Implementation details are provided in the Section B.

## 5 EXPERIMENTS

The goal of our experiments is to evaluate how effectively our method enables the Q-function to interpolate between underestimation and overestimation for out-of-distribution (OOD) actions, thereby improving overall policy performance.

### 5.1 BASELINES AND EXPERIMENTAL SETUP

We adopt the Warm Start RL (WSRL) framework (Zhou et al., 2024) as the backbone of our experimental setup. Specifically, we use CalQL (Nakamoto et al., 2023) for offline pretraining and Soft Actor-Critic (SAC) (Haarnoja et al., 2018) with an ensemble of 10 Q-networks for online fine-tuning. We evaluate our method against several baselines: We compare against the following methods: WSRL (Zhou et al., 2024), which performs a warmup phase at the start of fine-tuning without retaining the offline dataset; CalQL, which addresses excessive conservatism in CQL (Kumar et al., 2020) through calibrated Q-learning; RLPD (Ball et al., 2023), which integrates offline datasets with online rollouts; and two representative offline RL algorithms, CQL and IQL (Kostrikov et al., 2021).

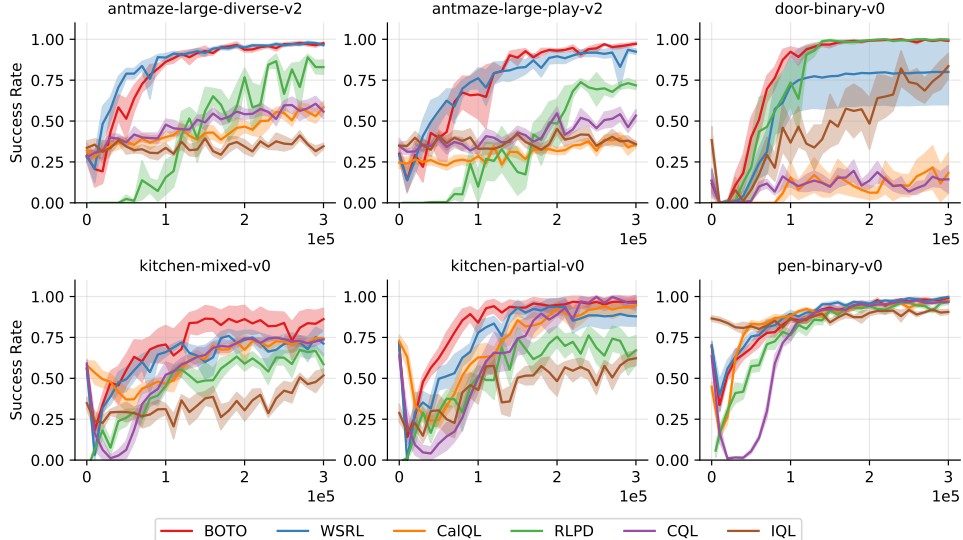

Figure 5: Performance comparison between our method and other baselines. Each plot shows the success rate over environments during online fine-tuning, with shaded regions indicating standard error across five random seeds. Ours achieves faster adaptation and higher final success rates than others in most tasks.

We train CalQL, CQL, and IQL during the offline pretraining phase and continue training them during the online phase while retaining access to the offline dataset. In contrast, the RLPD baseline is trained from scratch solely during the fine-tuning phase.

**RL tasks and offline datasets.** We evaluate our method on several widely used benchmark tasks and datasets, following prior work (Zhou et al., 2024; Shin et al., 2025). Specifically, we consider the Antmaze, Kitchen, and Adroit domains from the D4RL benchmark suite (Fu et al., 2020). The `Antmaze` domain includes long-horizon navigation tasks with sparse rewards, where an Ant robot must reach designated goals (`antmaze-large-diverse-v2`, `antmaze-large-play-v2`). The `Kitchen` environment involves executing a sequence of four subtasks using a Franka robotic arm in a simulated kitchen setting (`kitchen-partial`, `kitchen-mixed`). The `Adroit` domain comprises two dexterous manipulation tasks performed by a five-fingered robotic hand: pen reorientation (`pen-binary`) and door opening (`door-binary`). Further details on the implementation and environments are provided in the Section B.

## 5.2 Effectiveness of Q-Value Initialization with BOTO

Figure 5 presents a comparison of our method, BOTO, against baseline approaches across six benchmark environments during online fine-tuning. In all tasks, BOTO outperforms the baselines by effectively mitigating unlearning, which typically causes a performance drop at the start of fine-tuning. As discussed previously, CalQL, CQL, and IQL fail to perform in our setting due to misestimated Q-values learned during offline pretraining. While CalQL and IQL demonstrate reduced sensitivity to distributional mismatch between offline and online data by maintaining access to the offline dataset during online training, their performance frequently plateaus and does not exhibit consistent asymptotic improvement across most environments. In contrast, CQL, which enforces a regularization that constrains the Q-function to remain close to in-distribution actions, suffers from pronounced unlearning despite retaining the offline data, indicating that overly conservative constraints can hinder effective policy refinement during online adaptation. We evaluated the performance using the success rate, which is also used in existing baseline algorithms.

RLPD, which is trained exclusively from scratch during the online phase, requires substantially more training steps to reach performance levels comparable to O2O RL methods. Despite the inherent recovery capabilities of online RL algorithms, RLPD exhibits a notably slow learning curve in sparse

Table 1: Comparison of different offline RL algorithms. The Initial step corresponds to $t = 0$, the Unlearning step to $t = 10,000$, and the Final step to $t = 300,000$. The task name is succinctly stated: `antmaze-large-diverse-v2` (A-d), `kitchen-partial-v0` (K-p), and `kitchen-mixed-v0` (K-m). We average each score and get standard error across five random seeds.

| Task | | Initial step | Unlearning step | Final step |
|---|---|---|---|---|
| A-d | BOTO+CalQL | $0.29\pm0.08$ | $0.21\pm0.25$ | $0.98\pm0.02$ |
| | BOTO+CQL | $0.29\pm0.05$ | $0.22\pm0.26$ | $0.96\pm0.04$ |
| K-p | BOTO+CalQL | $0.70\pm0.05$ | $0.14\pm0.10$ | $0.97\pm0.08$ |
| | BOTO+CQL | $0.76\pm0.08$ | $0.10\pm0.13$ | $0.95\pm0.11$ |
| K-m | BOTO+CalQL | $0.56\pm0.08$ | $0.18\pm0.11$ | $0.86\pm0.15$ |
| | BOTO+CQL | $0.57\pm0.06$ | $0.19\pm0.11$ | $0.88\pm0.14$ |

reward environments such as `Antmaze` and `Kitchen`. This underscores the critical importance of initializing a well-trained Q-function to facilitate efficient and stable learning.

Although WSRL performs comparably to BOTO on `Antmaze` and `Adroit-pen`, its performance degrades significantly in `Kitchen` and `Adroit-door`. In particular, WSRL exhibits high performance variance across random seeds and fails to attain consistent asymptotic improvement in `Adroit-door`. In contrast, BOTO demonstrates enhanced robustness to unlearning and achieves faster recovery during the early stages of fine-tuning (around 10k steps) across most environments, highlighting the effectiveness of its Q-function initialization learned during the prior pretraining phase. Moreover, BOTO remains resilient to distributional mismatch, as the Q-function is trained to provide bounded estimates across the entire action space through regularization, enabling effective extrapolation beyond the support of the offline dataset and mitigating both excessive underestimation and overestimation for OOD actions.

## 5.3 COMPATIBILITY WITH OFFLINE RL ALGORITHMS

To further validate the generality of our approach, we evaluate BOTO using an alternative offline RL algorithm. This experiment demonstrates that BOTO effectively bounds the Q-function irrespective of the underlying pretrained policy and Q-function. As shown in Table 1, BOTO consistently achieves strong performance across different offline RL initializations, including both CalQL (BOTO+CalQL) and CQL (BOTO+CQL), underscoring its robustness and effectiveness independent of the choice of offline algorithm. We compare performance on three tasks at three key points: the onset of online fine-tuning (Initial step), the point at which unlearning occurs (Unlearning step), and the end of online fine-tuning (Final step). Our method reduces the unlearning effects commonly observed in both CalQL and CQL, resulting in more stable and higher overall performance. Specifically, in the `Kitchen` environments, BOTO offers a strong initialization that enables fast adaptation and leads to superior asymptotic performance.

## 5.4 CONCLUSION

In this work, we address a key bottleneck in offline-to-online reinforcement learning (O2O RL): the inaccurate estimation of state-action values for out-of-distribution actions inherited from the offline phase. To mitigate this issue, we propose an algorithm-agnostic method that adjusts the Q-function during online fine-tuning through a regularization scheme. Specifically, we introduce structured noise into action samples to explicitly bound Q-value estimates across the action space. This approach enables a flexible and principled initialization strategy that balances optimism and conservatism in Q-value estimation, facilitating rapid online adaptation. While selecting an appropriate value for the bias controller may require tuning for specific environments, empirical results across standard O2O RL benchmarks show that our method consistently improves both the stability and final performance of online fine-tuning, outperforming strong baselines. These findings highlight the critical role of Q-function initialization in O2O RL and point to promising future directions for robust policy adaptation under distributional shift.

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

# A THEOREM

## A.1 $\alpha$-NAMDP OBJECTIVE

*Proof.* To derive the optimal Q-function for the $\alpha$-NAMDP, we apply the Euler equation for functionals (Gelfand et al., 2000). Consider a functional of the following form:

$$J[u] = \int \cdots \int_{\mathbb{R}} F(x_1, \ldots, x_n, u) \, dx_1 \cdots dx_n \tag{4}$$

which depends on $n$ independent variables $x_1, \ldots, x_n$ and an unknown function $u$ defined over these variables. For the functional to attain an extremum, the following condition must be satisfied:

$$F_u(x) = 0 \quad \text{for all} \quad x \tag{5}$$

The $\alpha$-NAMDP objective Equation (3) can be cast in the same functional form:

$$J[Q] = \mathbb{E}_{\substack{a \sim p_D(\cdot|s) \\ \bar{a} \sim q_\sigma(\cdot|a)}} \left[ \left\| Q(s,\bar{a}) - \mathbb{E}_{\substack{s' \sim p_D(\cdot|s,a) \\ \tilde{a} \sim \pi(\cdot|s')}} \left[ R(s,a) - \alpha \|a - \bar{a}\|_2^2 + \gamma Q^\pi(s',\tilde{a}) \right] \right\|_2^2 \right]$$

$$= \int_{\mathbb{R}^n} \int_{\mathcal{A}} p_D(a|s) q_\sigma(\bar{a}|a)$$

$$\times \left\| Q(s,\bar{a}) - \mathbb{E}_{\substack{s' \sim p_D(\cdot|s,a) \\ \tilde{a} \sim \pi(\cdot|s')}} \left[ R(s,a) - \alpha \|a - \bar{a}\|_2^2 + \gamma Q^\pi(s',\tilde{a}) \right] \right\|_2^2 \, da \, d\bar{a}$$

$$= \int_{\mathbb{R}^n} F(\bar{a}_1, \ldots, \bar{a}_n, Q) \, d\bar{a}$$

where

$$F(\bar{a}_1, \ldots, \bar{a}_n, Q) = \int_{\mathcal{A}} p_D(a|s) q_\sigma(\bar{a}|a) \left\| Q(s,\bar{a}) - \mathbb{E}_{\substack{s' \sim p_D(\cdot|s,a) \\ \tilde{a} \sim \pi(\cdot|s')}} \left[ R(s,a) - \alpha \|a - \bar{a}\|_2^2 + \gamma Q^\pi(s',\bar{a}) \right] \right\|_2^2 \, da$$

Suppose $Q^*$ minimizes the $\alpha$-NAMDP objective equation 3. Then, by the Euler equation for functionals, we have:

$$F_{Q^*}(\bar{a}) = 2 \int_{\mathcal{A}} p_D(a|s) q_\sigma(\bar{a}|a) \left( Q^*(s,\bar{a}) - \mathbb{E}_{\substack{s' \sim p_D(\cdot|s,a) \\ \tilde{a} \sim \pi(\cdot|s')}} \left[ R(s,a) - \alpha \|a - \bar{a}\|_2^2 + \gamma Q^\pi(s',\tilde{a}) \right] \right) \, da = 0 \tag{6}$$

Therefore, it follows that:

$$\int_{\mathcal{A}} p_D(a|s) q_\sigma(\bar{a}|a) Q^*(s,\bar{a}) \, da$$

$$= \int_{\mathcal{A}} p_D(a|s) q_\sigma(\bar{a}|a) \mathbb{E}_{\substack{s' \sim p_D(\cdot|s,a) \\ \tilde{a} \sim \pi(\cdot|s')}} \left[ R(s,a) - \alpha \|a - \bar{a}\|_2^2 + \gamma Q^\pi(s',\tilde{a}) \right] \, da \tag{7}$$

This implies:

$$Q^*(s,\bar{a}) = \frac{\int_{\mathcal{A}} p_D(a|s) q_\sigma(\bar{a}|a) \mathbb{E}_{\substack{s' \sim p_D(\cdot|s,a) \\ \tilde{a} \sim \pi(\cdot|s')}} \left[ R(s,a) - \alpha \|a - \bar{a}\|_2^2 + \gamma Q^\pi(s',\tilde{a}) \right] \, da}{\int_{\mathcal{A}} p_D(a|s) q_\sigma(\bar{a}|a) \, da} \tag{8}$$

$$= \frac{\int_{\mathcal{S}} \int_{\mathcal{A}} p_D(a|s) q_\sigma(\bar{a}|a) p_D(s'|s,a) \mathbb{E}_{\tilde{a} \sim \pi(\cdot|s')} \left[ R(s,a) - \alpha \|a - \bar{a}\|_2^2 + \gamma Q^\pi(s',\tilde{a}) \right] \, da \, ds'}{\int_{\mathcal{A}} p_D(a|s) q_\sigma(\bar{a}|a) \, da} \tag{9}$$

$$= \int_{\mathcal{S}} \int_{\mathcal{A}} p_D(s'|s,a) p_D(\bar{a}|s,a,\sigma) \mathbb{E}_{\tilde{a} \sim \pi(\cdot|s')} \left[ R(s,a) - \alpha \|a - \bar{a}\|_2^2 + \gamma Q^\pi(s',\tilde{a}) \right] \, da \, ds' \tag{10}$$

We can separate this into two terms:

$$Q^*(s, \bar{a}) = \underbrace{\int_{\mathcal{A}} p_D(\bar{a}|s, a, \sigma) \left(R(s, a) - \alpha \|a - \bar{a}\|_2^2\right) da}_{R_\sigma(s, \bar{a})}$$

$$+ \gamma \underbrace{\int_{\mathcal{S}} \left(\mathbb{E}_{\tilde{a} \sim \pi(\cdot|s')} Q^\pi(s', \tilde{a})\right) \left(\int_{\mathcal{A}} p_D(s'|s, a) p_D(\bar{a}|s, a, \sigma) da\right) ds'}_{\mathbb{E}_{s' \sim P_\sigma(\cdot|s, \bar{a})}[\cdot]} \quad (11)$$

Hence, $Q^*$ satisfies the Bellman equation in the $\alpha$-NAMDP setting and represents the $Q$-value of policy $\pi$ under the $\alpha$-NAMDP dynamics. $\qquad \square$

## A.2 Q-VALUE BOUND IN THE $\alpha$-NAMDP

Let $\theta^*$ denote the parameters of $Q_\theta$ that minimize the functional objective defined in Equation (3), assuming a sufficiently expressive parameterization of the Q-function. Let $V^\pi$ be the true state value function under policy $\pi$ in the $\alpha$-NAMDP. Then, for any policy $\pi$, the learned Q-function $Q_{\theta^*}$ satisfies:

$$L^\pi(s, a) \leq Q_\theta(s, a) \leq U^\pi(s, a) \quad \forall(s, a) \in \mathcal{S} \times \mathcal{A}$$

where

$$L^\pi(s, a) = \min(0, -\alpha) \cdot \int_{\mathcal{A}} p_{\mathcal{D}}(\bar{a} \mid s, a, \sigma) \|a - \bar{a}\|_2^2 \, da + \inf_{a \in \mathrm{supp}(p_{\mathcal{D}}(\cdot|s))} \left[R(s, a) + \gamma \int_{\mathcal{S}} p_{\mathcal{D}}(s' \mid s, a) V^\pi(s') \, ds'\right],$$

$$U^\pi(s, a) = \max(0, -\alpha) \cdot \int_{\mathcal{A}} p_{\mathcal{D}}(\bar{a} \mid s, a, \sigma) \|a - \bar{a}\|_2^2 \, da + \sup_{a \in \mathrm{supp}(p_{\mathcal{D}}(\cdot|s))} \left[R(s, a) + \gamma \int_{\mathcal{S}} p_{\mathcal{D}}(s' \mid s, a) V^\pi(s') \, ds'\right].$$

*Proof.* Applying the Bellman equation for any state-action pair $(s, a) \in \mathcal{S} \times \mathcal{A}$, we have:

$$Q^\pi(s, a) = R_\sigma(s, a) + \gamma \int_{\mathcal{S}} P_\sigma(s' \mid s, \bar{a}) V^\pi(s') \, ds' \quad (12)$$

$$= \int_{\mathcal{A}} p_{\mathcal{D}}(a \mid s, \bar{a}, \sigma) \left[R(s, \bar{a}) - \alpha \|a - \bar{a}\|_2^2 + \gamma \int_{\mathcal{S}} p_{\mathcal{D}}(s' \mid s, \bar{a}) V^\pi(s') \, ds'\right] d\bar{a} \quad (13)$$

$$= \int_{\mathcal{A}} p_{\mathcal{D}}(a \mid s, \bar{a}, \sigma) \left[R(s, \bar{a}) + \gamma \int_{\mathcal{S}} p_{\mathcal{D}}(s' \mid s, \bar{a}) V^\pi(s') \, ds'\right] d\bar{a} - \alpha \int_{\mathcal{A}} p_{\mathcal{D}}(\bar{a} \mid s, a, \sigma) \|a - \bar{a}\|_2^2 \, d\bar{a}. \quad (14)$$

Since the dataset distribution $p_{\mathcal{D}}(\bar{a} \mid s, \bar{a}, \sigma)$ is supported only on $\mathrm{supp}(p_{\mathcal{D}}(\cdot \mid s))$ and integrates to 1, we bound the first integral term as follows:

$$\inf_{a \in \mathrm{supp}(p_{\mathcal{D}}(\cdot|s))} \left[R(s, a) + \gamma \int_{\mathcal{S}} p_{\mathcal{D}}(s' \mid s, a) V^\pi(s') \, ds'\right] \quad (15)$$

$$\leq \int_{\mathcal{A}} p_{\mathcal{D}}(a \mid s, \bar{a}, \sigma) \left[R(s, \bar{a}) + \gamma \int_{\mathcal{S}} p_{\mathcal{D}}(s' \mid s, \bar{a}) V^\pi(s') \, ds'\right] d\bar{a} \quad (16)$$

$$\leq \sup_{a \in \mathrm{supp}(p_{\mathcal{D}}(\cdot|s))} \left[R(s, a) + \gamma \int_{\mathcal{S}} p_{\mathcal{D}}(s' \mid s, a) V^\pi(s') \, ds'\right]. \quad (17)$$

Similarly, we bound the second integral term as follows:

$$\min(0, -\alpha) \cdot \int_{\mathcal{A}} p_{\mathcal{D}}(a \mid s, \bar{a}, \sigma) \|a - \bar{a}\|_2^2 \, d\bar{a} \quad (18)$$

$$\leq -\alpha \int_{\mathcal{A}} p_{\mathcal{D}}(\bar{a} \mid s, a, \sigma) \|a - \bar{a}\|_2^2 \, da \quad (19)$$

$$\leq \max(0, -\alpha) \cdot \int_{\mathcal{A}} p_{\mathcal{D}}(a \mid s, \bar{a}, \sigma) \|a - \bar{a}\|_2^2 \, d\bar{a} \quad (20)$$

Combining the two established inequalities, we finally derive:

$$\inf_{a \in \text{supp}(p_{\mathcal{D}}(\cdot|s))} \left[ R(s,a) + \gamma \int_{\mathcal{S}} p_{\mathcal{D}}(s' \mid s, a) V^{\pi}(s') \, ds' \right] + \min(0, -\alpha) \cdot \int_{\mathcal{A}} p_{\mathcal{D}}(a \mid s, \bar{a}, \sigma) \|a - \bar{a}\|_2^2 \, d\bar{a}$$

$$\leq Q^{\pi}(s,a) = Q_{\theta}(s,a) \quad \text{(by Theorem 1)}$$

$$\leq \sup_{a \in \text{supp}(p_{\mathcal{D}}(\cdot|s))} \left[ R(s,a) + \gamma \int_{\mathcal{S}} p_{\mathcal{D}}(s' \mid s, a) V^{\pi}(s') \, ds' \right] + \max(0, -\alpha) \cdot \int_{\mathcal{A}} p_{\mathcal{D}}(a \mid s, \bar{a}, \sigma) \|a - \bar{a}\|_2^2 \, d\bar{a}$$

This concludes the proof. □

# B    EXPERIMENTAL DETAILS

All experiments were conducted on a single NVIDIA RTX 3090 GPU and an Intel Xeon Gold 6330 CPU with 256 GB of RAM, running Ubuntu 22.04.4 LTS and JAX version 0.6.1.

Our code is available at: https://anonymous.4open.science/r/BOTO-5FF8/README.md

In all experiments, we selected the value of $\alpha$ from the range $[-1, 1]$ with a discretization interval of 0.1.

| Task | $\alpha$ | BOTO step |
|---|---|---|
| antmaze-large-diverse-v2 | -1.0 | 100,000 |
| antmaze-large-play-v2 | -0.7 | 100,000 |
| kitchen-partial-v0 | 0.8 | 250,000 |
| kitchen-mixed-v0 | 1.0 | 250,000 |
| door-binary-v0 | 0.0 | 100,000 |
| pen-binary-v0 | -0.9 | 100,000 |

Table 2: Hyperparameters: $\alpha$ and BOTO training step per task.

## B.1    NOISE DISTRIBUTION

We also, to sample the perturbed action, follow the Hybrid Noise Distribution proposed by Oh & Lee (2025):

$$q_\sigma^{\text{hyb}}(\bar{a} \mid a) = \mathbb{E}_{\lambda \sim \mathcal{U}(\log \sigma, 0)} \left[ q_{\exp(\lambda)}^{\mathcal{A}}(\bar{a} \mid a) \right],$$

where

$$q_t^{\mathcal{A}}(\bar{a} \mid a) = \alpha(t)\, \mathcal{U}(\bar{a} \mid \mathcal{A}) + (1 - \alpha(t))\, \mathcal{N}(\bar{a} \mid a, t^2 \mathbf{I})$$

We set $\log \sigma = -30$ for `antmaze-large-diverse-v2`, and $\log \sigma = -20$ for all other environments.

## B.2    ENVIRONMENT DETAILS

We evaluate our method across several standard benchmark tasks and datasets widely adopted in previous studies (Zhou et al., 2024; Shin et al., 2025). Our evaluation spans the Antmaze, Kitchen and Adroit domains from the D4RL benchmark suite (Fu et al., 2020). (1) The `Antmaze` domain, comprising long-horizon navigation challenges with sparse rewards, where an 8-DoF Ant robot must reach predefined goals (`antmaze-large-diverse-v2`, `antmaze-large-play-v2`). (2) The `Kitchen` environment, a long-horizon task where a 9-DoF Franka arm must execute a sequence of four subtasks in a simulated kitchen setting (`kitchen-partial` and `kitchen-mixed`). (3) Two `Adroit` manipulation tasks involving a 28-DoF five-fingered robotic hand: reorienting a pen to a target pose (`pen-binary`) and opening a door by manipulating a latch (`door-binary`).

## B.3    COMPUTE RESOURCE

Table 3 summarizes the wall-clock training time across different tasks.

| Task | Time |
|---|---|
| antmaze-large-diverse-v2 | 2h 48m |
| antmaze-large-play-v2 | 2h 40m |
| kitchen-partial-v0 | 5h 53m |
| kitchen-mixed-v0 | 4h 51m |
| door-binary-v0 | 5h 49m |
| pen-binary-v0 | 4h 45m |

Table 3: Training Wall-Clock Time

