# OpenReview forum: "Explicitly Bounding Q‑Function Estimates for Offline-to-Online Reinforcement Learning"
_ICLR.cc/2026/Conference — ICLR 2026 Conference Withdrawn Submission_

### Official Review · Reviewer_5yeW · 2025-10-28

**Soundness:** 2
**Presentation:** 2
**Contribution:** 2
**Rating:** 2
**Confidence:** 3

**Summary:**

This paper introduces a new method, BOTO, for offline-to-online (O2O) reinforcement learning. To address the problem of Q-value misestimation due to OOD actions, the authors propose to explicitly bound Q-value estimates across the entire action space, which is achieved by injecting structured noise into the dataset actions. This process mitigates both overestimation and underestimation and improves the performance on standard O2O RL benchmarks.

**Strengths:**

- Q-value misestimation is a well-known limitation of O2O fine-tuning. The motivation is clear and the proposed $\alpha$-NAMDP makes sense.
- The results look good, the proposed method performs more stable compared to other baselines.

**Weaknesses:**

- The idea of injecting noise into the dataset actions is native. An illustration of how it mitigates Q-value misestimation should be provided, e.g., through a toy example or some ablation experiments.
- The tunable parameter $\alpha$ is so important and sensitive to this framework. I found from the Table 2 that you choose different $alpha$ values for different tasks using grid research (from -1 to 1), which looks like this parameter is over-tuned. It would be good to have a detailed analysis or ablation study on the $\alpha$.
- The model is evaluated on few D4RL benchmarks. However, as in the CQL paper, the proposed method should also be evaluated on diverse datasets of the same task. For example, 1) the "-random", "-expert" and "-medium" in Gym domains; 2) "-umaze", "-medium", and "-large" settings in AntMaze.
- The ablation experiments on warmup phase is missing.

**Questions:**

- How do you get the illustration of Q-value misestimation cases in Figure 1 (from a real task or a synthetic dataset)? It would be better to provide more details.

---

### Official Review · Reviewer_66d1 · 2025-10-31

**Soundness:** 3
**Presentation:** 3
**Contribution:** 3
**Rating:** 6
**Confidence:** 3

**Summary:**

This paper addresses Q-value misestimation in Offline-to-Online Reinforcement Learning (O2O RL) by proposing BOTO (Bounding Q-function Estimates for Offline to Online RL), an algorithm-agnostic method that explicitly bounds Q-value estimates across the entire action space. The core contribution is a regularization approach that injects structured noise into dataset actions during Q-learning prior to online fine-tuning. The authors formalize their approach through the α-Noisy Action MDP (α-NAMDP) framework, showing that their method corresponds to Q-learning in a modified MDP with a tunable bias parameter α that controls the degree of conservatism versus optimism in Q-value estimates. The method consists of three stages: (1) offline pre-training using standard offline RL algorithms, (2) BOTO training that optimizes the Q-function using action noise injection with a warmup dataset, and (3) online fine-tuning using standard online RL algorithms. Empirical evaluations on D4RL benchmarks (Antmaze, Kitchen, and Adroit domains) demonstrate that BOTO outperforms strong baselines including WSRL, CalQL, RLPD, CQL, and IQL by mitigating both underestimation and overestimation issues during the online fine-tuning phase.

**Strengths:**

Principled theoretical framework:
The paper provides rigorous theoretical analysis by introducing the α-NAMDP formulation with two key theorems: Theorem 1 establishes equivalence between minimizing the proposed objective and Q-learning in the α-NAMDP, while Theorem 2 derives provable bounds on Q-values across the entire action space, including out-of-distribution (OOD) actions. This theoretical grounding distinguishes the work from heuristic approaches and provides formal guarantees on the boundedness of learned Q-functions.

Clear problem motivation and analysis:
The paper systematically analyzes the failure modes of Q-value misestimation in O2O RL, distinguishing between global underestimation, local underestimation, and overestimation scenarios. Figure 1 effectively illustrates these failure modes, and the discussion clearly articulates why existing methods suffer during online fine-tuning. This thorough problem characterization strengthens the motivation for the proposed solution.

Algorithm-agnostic design:
BOTO is designed to work with any offline RL algorithm, which is demonstrated empirically through experiments combining BOTO with both CalQL and CQL as base methods. Table 1 shows consistent improvements regardless of the underlying offline algorithm, highlighting the generality and practical applicability of the approach. This flexibility is valuable for practitioners who may have existing offline RL pipelines.

Comprehensive experimental evaluation:
The empirical evaluation covers six benchmark tasks across multiple domains (Antmaze, Kitchen, Adroit) and compares against five strong baselines. The results consistently demonstrate that BOTO achieves faster adaptation and higher final performance, particularly in mitigating the "unlearning" phenomenon that causes performance drops at the onset of fine-tuning. The inclusion of standard error bars across five random seeds strengthens the reliability of the results.

Effective visualization of the bounding mechanism:
Figure 2 provides compelling visual evidence of how BOTO produces bounded Q-values across the action space compared to baseline methods (CQL, CalQL, IQL), clearly showing how different α values modulate the degree of conservatism and optimism. Figure 3 further validates the theoretical bounds derived in Theorem 2 against empirical results in a controlled bandit setting.

**Weaknesses:**

Limited novelty over prior work. The paper builds heavily on Oh & Lee (2025), extending the Noisy Action MDP to α-NAMDP by introducing a tunable parameter α. While this generalization provides useful control over bias, the core mechanism of action noise injection for Q-value bounding is not novel. The incremental nature of the contribution relative to the base NAMDP framework should be more explicitly acknowledged, and the paper would benefit from a more detailed comparison highlighting what specific advantages the α parameter provides beyond the original formulation.

Hyperparameter sensitivity not thoroughly addressed. The paper acknowledges that "selecting an appropriate value for the bias controller may require tuning for specific environments", but provides insufficient guidance on how to set α in practice. Table 2 shows that optimal α values vary substantially across tasks (ranging from -1.0 to 1.0), yet the paper lacks principled strategies for hyperparameter selection. The discretization interval of 0.1 across the range [-1, 1] requires evaluating 21 different values, which could be computationally expensive. A sensitivity analysis or adaptive selection method would strengthen the practical applicability.

Insufficient ablation studies. The paper lacks important ablation studies to isolate the contributions of different components. Specifically: (1) What is the individual contribution of the warmup phase versus the BOTO training objective? (2) How sensitive is performance to the warmup dataset size (Dwarmup)? (3) What is the impact of different noise distributions beyond the hybrid noise distribution adopted from Oh & Lee (2025)? (4) How does the BOTO training duration affect results? While Table 2 provides some hyperparameter values, systematic ablation studies would clarify which design choices are critical.

Limited theoretical insights on α selection. While Theorems 1 and 2 characterize the learned Q-function under α-NAMDP, they do not provide guidance on how to choose α for a given task. The bounds in Theorem 2 depend on the reward function and value function, but it's unclear how these quantities can be estimated a priori to inform α selection. Developing a theoretical connection between task characteristics (e.g., sparsity of rewards, dataset quality) and optimal α values would significantly enhance the practical utility of the framework.

Missing comparisons with recent methods. While the paper compares against several strong baselines, it omits comparisons with other recent O2O RL methods that also address Q-value estimation issues, such as SO2 (Zhang et al., 2024; https://arxiv.org/pdf/2312.07685) which uses perturbed value updates, or ENOTO (Zhao et al., 2024; https://arxiv.org/pdf/2306.06871) which employs Q-ensembles. Including these comparisons would better position BOTO within the current landscape of O2O RL research.

**Questions:**

Q1: How should α be selected in practice? Could the authors provide more principled guidance or an adaptive method for selecting α? For instance, could α be estimated based on properties of the offline dataset (e.g., coverage, quality, diversity) or initial Q-value statistics?

Q2: How does BOTO perform on suboptimal or mixed-quality datasets? All experiments use standard D4RL datasets. How does BOTO behave when the offline dataset is of very poor quality or contains multi-modal behavior policies? Does the method provide robustness advantages in these more challenging scenarios?

Q3: Why does BOTO use the offline dataset in addition to warmup data during BOTO training? The paper constructs DBOTO = D_offline ∪ D_warmup, but it's unclear why the offline dataset is retained during the BOTO training phase. What would happen if BOTO training used only D_warmup? Does this design choice relate to preventing distributional shift or maintaining coverage?

Q4: Can the bounds in Theorem 2 be tightened? The bounds depend on the infimum and supremum over the support of the dataset distribution. In practice, are these bounds tight, or is there significant slack? Could empirical analysis on the benchmark tasks characterize how tight the bounds are in practice?

Q5: How does BOTO compare to ensemble-based methods? Methods like ENOTO use Q-ensembles to address similar issues in O2O RL. What are the relative advantages of the noise injection approach versus ensemble methods in terms of performance, computational cost, and implementation complexity?

---

### Official Review · Reviewer_yUt6 · 2025-10-31

**Soundness:** 3
**Presentation:** 3
**Contribution:** 1
**Rating:** 2
**Confidence:** 4

**Summary:**

The paper presents an algorithm-agnostic approach to address challenges in the O2O reinforcement learning paradigm. In O2O RL, agents are pretrained on static datasets and refined through limited online interactions, but this process is hindered by Q-value misestimations for OOD actions, leading to underestimation or overestimation that destabilizes fine-tuning. The proposed method, BOTO, injects structured action noise during a pre-fine-tuning phase to regularize Q-values across the entire action space. This is formalized via α-NAMDP, where a tunable parameter α balances conservatism and optimism in Q-value bounds.

Key contributions include:
- Introduction of a Q-value bounding technique through action noise injection that explicitly regularizes OOD actions while mitigating both underestimation and overestimation.
- Theoretical analysis establishing equivalence to Q-learning in the α-NAMDP and deriving explicit bounds on Q-value estimates.
- Empirical demonstrations on standard O2O RL benchmarks.

**Strengths:**

1. The writing is logically coherent. A feasible improvement plan is proposed to address the inaccuracy of Q estimation of OOD actions.
2. The BOTO algorithm seems to achieve state-of-the-art performance in experiments compared to other algorithms.
3. The author constructed a reliable theoretical framework for this algorithm: α-NAMDP and proved the global boundedness of Q under this framework.

**Weaknesses:**

A substantive assessment of the weaknesses of the paper. Focus on constructive and actionable insights on how the work could improve towards its stated goals. Be specific, avoid generic remarks. For example, if you believe the contribution lacks novelty, provide references and an explanation as evidence; if you believe experiments are insufficient, explain why and exactly what is missing, etc.

1. This study appears to have only introduced a hyperparameter from previous research[1], which severely undermines its contribution.
2. The author does not seem to have designed experiments separately to verify the performance of policies in the O2O process, although the author repeatedly emphasizes that their method affects this process.
3. At present, the scope of this algorithm seems to be limited to the O2O process, and it does not appear to benefit subsequent online RL.

[1]JunHyeok Oh and Byung-Jun Lee. Offline reinforcement learning with penalized action noise injection. arXiv preprint arXiv:2507.02356, 2025.

**Questions:**

1. Apart from introducing hyperparameters, could you further clarify the difference between this study and [1]?
2. Can the experiment contain the entire process of policy performance (Offline Pre-training, O2O, Online fine-tuning)?
3. Directly performing online pre-training on Q also seems feasible. How does this technique compare with online pre-training? Training time or computational cost?

[1]JunHyeok Oh and Byung-Jun Lee. Offline reinforcement learning with penalized action noise injection. arXiv preprint arXiv:2507.02356, 2025.

---

### Official Review · Reviewer_cHkV · 2025-11-01

**Soundness:** 2
**Presentation:** 2
**Contribution:** 2
**Rating:** 4
**Confidence:** 2

**Summary:**

The paper targets the Offline-to-Online (O2O) RL setting, where value misestimation for out-of-distribution (OOD) actions inherited from the offline phase can derail early online fine-tuning. It proposes BOTO, which injects structured noise into actions while adjusting the Bellman targets to explicitly bound Q-values across the entire action space. A tunable parameter αtrades off optimism vs. conservatism, yielding an α-Noisy Action MDP interpretation with theoretical bounds on Qfor both in-distribution and OOD actions. Empirically, the method improves stability and final performance on D4RL domains (AntMaze, Kitchen, Adroit) under a Warm Start RL protocol.

**Strengths:**

The paper formalizes noise-perturbed Q-learning targets and proves equivalence to learning in an α-NAMDP, including upper/lower bounds on Q that cover OOD actions


The single parameter αgives a transparent way to bias OOD estimates and thereby shape early online behavior. Figure 3 and Theorem 2 illustrate the induced bounds.

On AntMaze/Kitchen/Adroit, learning curves and Table 1 show faster adaptation and stronger final success rates vs. CalQL, CQL, IQL, RLPD, and WSRL.

**Weaknesses:**

Limited novelty. The core idea closely follows and extends Offline RL with Penalized Action Noise Injection (Oh & Lee, 2025) by adding the αbias control; the paper should more sharply articulate what is genuinely new in analysis or practice beyond that prior.

The conclusion acknowledges that choosing the bias controller may require per-environment tuning, which could undercut the algorithm-agnostic appeal.

**Questions:**

In what ways (theory or practice) does BOTO substantively differ from penalized action-noise injection beyond introducing the $\alpha$ controller and the $\alpha$-NAMDP lens?

---

### Note · Authors · 2025-11-12

I have read and agree with the venue's withdrawal policy on behalf of myself and my co-authors.